# Biochemical, Genetic and Clinical Diagnostic Approaches to Autism-Associated Inherited Metabolic Disorders

**DOI:** 10.3390/genes14040803

**Published:** 2023-03-27

**Authors:** Udara D. Senarathne, Neluwa-Liyanage R. Indika, Aleksandra Jezela-Stanek, Elżbieta Ciara, Richard E. Frye, Cliff Chen, Karolina M. Stepien

**Affiliations:** 1Department of Biochemistry, Faculty of Medical Sciences, University of Sri Jayewardenepura, Nugegoda 10250, Sri Lanka; 2Department of Chemical Pathology, Monash Health Pathology, Monash Health, Melbourne, VIC 3168, Australia; 3Department of Genetics and Clinical Immunology, National Institute of Tuberculosis and Lung Diseases, 01-138 Warsaw, Poland; 4Department of Medical Genetics, The Children’s Memorial Health Institute, 04-730 Warsaw, Poland; 5Autism Discovery and Treatment Foundation, Phoenix, AZ 85050, USA; 6Clinical Neuropsychology Department, Manchester Centre for Clinical Neurosciences, Salford Royal NHS Foundation Trust, Salford M6 8HD, UK; 7Adult Inherited Metabolic Diseases, Mark Holland Unit, Salford Royal NHS Foundation Trust, Salford M6 8HD, UK; 8Division of Diabetes, Endocrinology and Gastroenterology, University of Manchester, Manchester M13 9PL, UK

**Keywords:** autism, ASD, WES, NGS, neuropsychology, inherited metabolic diseases

## Abstract

Autism spectrum disorders (ASD) are a heterogeneous group of neurodevelopmental disorders characterized by impaired social interaction, limited communication skills, and restrictive and repetitive behaviours. The pathophysiology of ASD is multifactorial and includes genetic, epigenetic, and environmental factors, whereas a causal relationship has been described between ASD and inherited metabolic disorders (IMDs). This review describes biochemical, genetic, and clinical approaches to investigating IMDs associated with ASD. The biochemical work-up includes body fluid analysis to confirm general metabolic and/or lysosomal storage diseases, while the advances and applications of genomic testing technology would assist with identifying molecular defects. An IMD is considered likely underlying pathophysiology in ASD patients with suggestive clinical symptoms and multiorgan involvement, of which early recognition and treatment increase their likelihood of achieving optimal care and a better quality of life.

## 1. Introduction

Autism spectrum disorders (ASD) are defined by the revised version of the Diagnostic and Statistical Manual of Mental Disorders-5 (DSM-5-TR) as neuro-developmental disorders characterized by persistent deficits in reciprocal social communication and social interaction (Criterion A) and restricted and/or repetitive patterns of behaviour, interests, or activities (Criterion B), present from early childhood (Criterion C) causing clinically significant impairment in social, occupational, or other important areas of current functioning (Criterion D). The symptoms under criteria A and B are not better explained by intellectual disability or global developmental delay (Criterion E). It is known as a “spectrum” disorder because there is wide variability in the severity and pattern of symptoms, progression of the disease, and prognosis. ASD is accompanied by varying degrees of intellectual impairment and is associated with many genetic, neurodevelopmental, mental, and behavioural disorders and environmental factors [1].

The global prevalence of ASD is estimated to be 0.4–1%, with American and European countries having a higher prevalence (1%) compared to Asian (0.4%) countries [2]. The biannual estimates by ASD and the Autism and Developmental Disabilities Monitoring (ADDM) Network report a two-fold increase in ASD prevalence among 8-year-old children in the United States during the past decade, from 2008 (1.1%) to 2018 (2.3%), with boys being four times more affected than girls [3].

The etiological factors and the proposed pathogenic mechanisms in ASD are intricate and involve the interaction of genetic, epigenetic, and environmental elements [4]. The main dysregulations, including intestinal dysbiosis [5], immune dysfunction, metabolic dysfunction [6], and metal dyshomeostasis [7], have been identified using molecular biomarkers, such as gut microbiome-related metabolites, cytokine profile, autoantibodies, metabolites, and vitamin and mineral profiles [8]. These are interconnected; hence, an ASD individual can exhibit several dysfunctions. For instance, a child with gut microbial dysbiosis may exhibit core symptoms of ASD along with immune dysfunction, altered microbial metabolites, and epigenetic changes [9,10]. The gut microbiome may modulate the gut–brain axis through microbiota-derived signalling molecules, immune mediators, gastrointestinal hormones, etc. [10,11].

Although each inherited metabolic disorder (IMD) is rare in isolation, collectively, all IMDs have a combined incidence of 1:800 to 1:2500 [12,13,14]. Contributing to the heterogeneity, a number of IMD, such as 22q11.2 deletion, Angelman syndrome, Cohen syndrome, Noonan syndrome, and fragile X syndrome, are known to associate with autistic symptoms, corroborating the evidence for genetic etiologies of ASD [15,16,17,18,19,20,21]. It should also be noted that the symptoms of ASD can be present in association with many IMDs [16]. The prevalence of IMDs among ASD individuals has been estimated in different studies, ranging from 0.7% to 2.7% [22,23,24]. However, the true prevalence of IMDs among ASD patients has been speculated to be higher [25], corroborating the finding of more than 30% of ASD individuals having some form of metabolic derangement by Spilioti et al. [24]. Furthermore, more than 50% of the IMDs present with neurodevelopmental symptoms, and ASD primarily being a neurodevelopmental disorder, the rational investigation of ASD individuals for probable IMDs is appropriate, especially in communities with a high level of consanguinity [26,27,28].

To date, more than 100 autism-risk genes have been identified with a genetic cause found in 10–20% of the cases of ASD during the investigation process [22,29]. Table 1 elaborates the IMDs associated with ASDs. A whole-exome sequencing (WES) study identified that 52% were nuclear sequence-level variants, 46% were nuclear structural variants, and 2% were mitochondrial variants [30]. Exome sequencing is identified as a first-tier clinical diagnostic test for individuals with neurodevelopmental disorders, including ASD, with a molecular diagnostic yield of 16% (CI: 11–24%) in ASD [31]. Additionally, chromosomal microarray (CMA) has revealed definitively pathogenic copy number variants (CNVs) in 5.4% to 14% of individuals with ASD [32]. Even though ASD-associated biallelic genetic variants are commonly reported in consanguineous families, some were observed in non-consanguineous families. Siblings from consanguineous and/or multiplex families who share identical homozygous biallelic mutation may show variable ASD phenotypes with or without intellectual disability, epilepsy, and other clinical features [33]. Moreover, it is estimated that de novo variants in protein-coding genes contribute to risk in approximately 30% of simplex families. At the same time, de novo variants in non-coding regions of the genome (particularly gene promoters) also contribute to ASD pathogenesis [34]. De novo variants in the non-coding region highlight the current need to perform ASD genetic studies using whole genome sequencing (WGS) instead of traditional exome studies [35].

Large population-based studies on the outcome of metabolic investigations in screening for IMDs among ASD individuals are lacking; hence, accurate prevalence and diagnostic yield estimates are not available. Despite the observed low yield of routine metabolic testing in ASD individuals, the positive impact on ASD management may be significantly high as it paves the path to better understanding the underlying pathophysiology, inheritance pattern, and availability of treatment [32].

With the advancement of tandem mass spectrometry, most developed countries employ expanded newborn screening (NBS) as an important public health strategy, though it is yet to become a priority in developing countries [75,76]. Therefore, children who receive an expanded NBS have the advantage of receiving the diagnosis of an IMD, if present, before they would develop any autistic symptoms, whereas, in developing countries, ASD patients may only undergo targeted/selective screening for IMDs following the establishment of the primary diagnosis of a neurological condition or ASD. Selective screening for IMDs in ASD subjects may be a cost-effective strategy to identify the patients with IMDs that remain undiagnosed in countries where mass NBS for IMDs is unavailable. If the diagnosis of an IMD was established, treatment targeted at specific metabolic abnormalities in children with ASD has potential benefits [24,77]. For instance, identifying mitochondrial dysfunction enables the implementation of targeted therapies such as vitamins, carnitine, Coenzyme Q10, etc. [78,79]. In addition, research on advancing medical treatments, such as hematopoietic stem cell transplantation, is underway, offering hope for IMDs associated with ASD [80,81].

Investigating for IMDs associated with ASD should take a multidisciplinary approach led by a clinician (psychiatrist/neurologist/paediatrician/metabolic physician) with a well-informed team, including a chemical pathologist, clinical biochemist, and clinical geneticist. The primary aim of this endeavour is to understand the underlying pathophysiology, to provide genetic counselling where appropriate, and to determine the potential therapies. The success of investigating for IMDs in ASD patients depends on the understanding of different phenotypes of known syndromes and IMDs that overlap with ASD and defining a tailored biochemical and molecular evaluation plan catering to the needs of the individual patient based on their unique clinical information [82]. The assays to be considered as a screening panel should be based on common IMDs associated with ASD, the local epidemiology of IMDs in a given country, and available resources. In addition, the coverage of the existing NBS program should be considered in determining redundant investigations. This review outlines a step-by-step approach to the investigations of ASD when a diagnosis of an IMD is suspected.

## 2. Biochemical Investigations

The main biochemical mechanisms proposed in ASD include mitochondrial dysfunction [78,83], oxidative stress [84], impaired methylation capacity [85], and altered amino acid metabolism [86]. Interestingly, ASD patients had these metabolic abnormalities in the brain regions involved in speech and auditory processing, social behaviour, sensory and motor coordination, and memory, the core symptoms of ASD [87]. Corroborating the presence of mitochondrial dysfunction in ASD, a 2020 study demonstrated atypical mitochondrial morphology with mitochondrial electron transport chain abnormalities in the fibroblasts of children with ASD [88]. Furthermore, several mitochondrial functional biomarkers, such as lactate, pyruvate, carnitine, and ubiquinone, are significantly altered in ASD, while some even correlate with severity [78,83]. The impaired methylation capacity is evident by the significantly reduced S-adenosylmethionine/S-adenosylhomocysteine ratio (methylation index) in ASD, while reduced methionine levels and increased homocysteine levels indicate the impaired remethylation of homocysteine to methionine [84]. The reduced availability of the cofactors for the remethylation pathway in the brain may be attributed to reduced blood folate [84], vitamin B12 levels [84], low-activity variants of the genes (*MTHFR*, *DHFR*, *FOLR1*) affecting folate and cobalamin metabolism [58,89], and cerebral folate deficiency due to folate receptor α autoantibodies [90]. Deth et al. proposed a “redox/methylation hypothesis of autism” describing the pathogenesis of oxidative stress, precipitated by environmental factors in genetically vulnerable individuals, which limits the activity of methionine synthase due to its dependency on cobalamin and folate, hence impaired methylation, including dopamine-stimulated phospholipid methylation [91]. Reduced methylation capacity may also mediate epigenetic changes through modulating DNA and histone methylation in a sex-dependent manner [58]. These mechanisms are thought to induce changes in the neurotransmitter systems, such as γ-aminobutyric acid (GABA) and glutamate, serotonin, dopamine, melatonin, and acetylcholine [92], resulting in core symptoms and co-occurring behavioural and neurological symptoms.

Pathway enrichment analysis has demonstrated that microRNAs (miRNAs) differently expressed in ASD are involved in metabolic pathways, such as steroid biosynthesis, fatty acid metabolism, lysine degradation, and biotin metabolism [93]. Correspondingly ASD-associated IMDs can be mapped to disorders of the metabolism of amino acids, carbohydrates, fatty acids, sterol biosynthesis, ketone bodies, creatine, vitamins (B12, folate, and biotin), cofactors, nucleotide, mitochondrial metabolite repair, etc. Table 1 summarizes the reported IMDs in ASD according to the international classification of IMDs [94]. Importantly, these genetic and metabolic disorders are associated with marked cognitive impairment and other clinical features, such as macrocephaly, extrapyramidal signs, motor developmental delay, dysmorphic features, failure to thrive, or hepatosplenomegaly, which are atypical for patients with ASD [17,42].

In the context of complex pathogenesis with multiple co-morbidities and the evidence of metabolic involvement and strong heritable component, although the pathophysiology is not well comprehended, the current disease trajectory is developing towards a better understanding of the metabolic component of ASD in order to advance future interventions to improve the overall quality of life for ASD individuals [95,96,97]. The future developments are mainly focused on improving early and accurate diagnostic algorithms in unravelling the metabolic and other components of ASD. The diagnostic yield of metabolic investigations in patients with isolated ASD and no clinical symptoms appears to be low [22,23].

Before the establishment of NBS, a large proportion of these individuals would have been brought to medical attention only after the development of autistic symptoms. However, with the NBS programs, especially in developed countries, many such individuals are brought to medical attention early, and necessary treatments are instigated; a good example is phenylalanine ketonuria, an IMD characterized by intellectual disability and ASD [52,63,98]. Therefore, the diagnostic approach to ASD should be equipped with a rational consideration of possible IMDs, as some are treatable [54]. A recently suggested approach to such investigation is untargeted metabolomic profiling, as many ASD patients demonstrate a wide range of metabolic abnormalities, from micronutrient deficiencies to severe metabolic derangements [99]. The basis for untargeted metabolic profiling is the emerging evidence of potential novel biomarkers of IMD associated with ASD, as found in many cohort studies conducted worldwide, further opening new avenues of treatments [24]. However, the additional healthcare costs need to be considered while expanding the investigation profile to weigh the medical benefits of further testing, and it may not be cost-effective in non-syndromic ASD [99]. Schiff et al. demonstrated that the prevalence of IMD among non-syndromic ASD individuals is not higher than the general population (<0.5%) by conducting a systematic metabolic work-up, highlighting the importance of a rational approach to metabolomics in ASD [22] (Table 1, Table 2 and Table 3).

## 3. Genetic Investigations

ASD frequently occurs in some types of IMDs, typically alongside some degree of developmental delay, intellectual disability, or motor impairment. The true prevalence of IMDs among ASD individuals could be higher than what is currently estimated due to the possible missed diagnosis of some cases [25]. Due to the observed evidence of heritability, research on the genetic aspects of ASD was observed to advance as early as the 1960s, which initially focused on structural DNA alterations, whereas the investigations are moving towards gene expression and epigenetics in recent years [114]. Advances and applications of genomic testing technology, together with large-scale projects (DDD study [115], 100,000 genome study [116]), have significantly improved the diagnostic yield of metabolic work-up in ASD individuals.

A gender bias has been observed historically in the incidence of neurodevelopmental disorders, with a four-to-five-fold higher incidence in males, which is more prominent in individuals with a milder degree of cognitive impairment [117,118]. Although this phenomenon would point toward the possibility of an association with X-linked genes, only 10% of the reported male excess can be attributed to X-linked genes. Further, ASD recurrent rates among the siblings of a proband are estimated to be 4% and 7%, for males and females, respectively, with the less frequent gender exhibiting a higher recurrence rate [114,119].

CMA allows the detection of clinically relevant chromosomal abnormalities in up to 10% of ASD individuals [98,116]. Similarly, exome sequencing has contributed to identifying rare variants in a significant number of ASD patients, as there is no clear definition with biochemical testing and imaging studies in all cases of IMDs. In fact, some cases are altogether missed unless genetic testing with high-throughput sequencing is performed, such as WES. WES enables the identification of genetic changes (Table 1), mainly within the coding sequences of all genes in the human genome. The analysis of the sequence of all genes, instead of individual candidate genes, significantly reduces diagnostic time and may also enable the discovery of new pathogenic variants. The significant development of NGS sequencing technology in recent years has dramatically reduced the cost of extensive analysis, making it feasible for its application in routine diagnostics. Normally, WES studies covering the exome regarding protein-coding exons and short adjacent regions (e.g., edges of introns) analyse only 1–2% of the genome (NCBI, NIH, USA). However, 85–89% of known pathogenic SNVs (single nucleotide variants; NCBI, NIH, USA) are located in this small study area [120]. Exome comprises only a small percentage of the genome, whereas the bulk of genes are introns, i.e., regions that do not encode protein sequences. Only 19,969 are identified as protein-coding genes out of the 63,494 genes identified as per the current annotation of the human genome (T2T-CHM13) [121].

Recent studies have also pointed toward the role of epigenetic dysregulation, mechanisms with no impact on the DNA sequence, in ASD pathogenesis [122]. Epigenetics can be defined as the regulation of gene expression by modulating chromatin formation [123], and reported instances of such dysregulations include impaired DNA methylation in ASD individuals [124,125], pathogenic variants in the *HIST1H1E* gene encoding H1 histone linker protein [126], activating immune responses during pregnancy increasing susceptibility to ASD [127], and alteration of the miRNA expression in ASD patients [128]. The WES identifies alterations in most protein-coding genes. Some centres also detect alterations in the remainder of genome encoding, for example, regulatory RNAs, or non-coding areas of the genome containing known pathogenic alterations. An autistic multigene panel sequencing analysis, including SNV and CNV detection, is recommended to identify pathogenic changes effectively. Considering the high diagnostic yield of WES in children with congenital anomalies, developmental delay, or intellectual disability, the American College of Medical Genetics and Genomics (ACMG) strongly recommends that WES/WGS be considered as a first-tier or second-tier test [129]. Its clinical utility can have a significant effect on long-term patient management and, when considered early in the diagnostic evaluation, may offer more cost-effective avenues of treatment in the long run. Furthermore, some genetic variants could be confirmed with biochemical evidence, such as defective enzyme level assessment or demonstrating the pathognomonic biochemical picture.

## 4. Clinical and Neuropsychology Assessments

The current recommendation is that further metabolic investigation should always be guided by detailed medical and family history and a physical examination to identify suggestive features, such as consanguinity, multiplex families, dysmorphic features, epilepsy, neurocutaneous manifestations, and existing metabolic derangements [82,98].

In some IMDs, some salient features of ASD may correspond to an underlying pathology, which often includes multiorgan involvement, such as in lysosomal storage disorders. Table 1 presents a list of IMDs associated with ASD. Only a few IMDs would be characterised by isolated ASD as the main clinical presentation, particularly at the time of the diagnosis. Schiff et al. (2011) confirmed in their study that routine metabolic screening would not add to the causative diagnosis in non-syndromic ASD [22]. Therefore, although the relationship between ASD and IMDs is unquestionable, it is likely to be confined to the subset of rare patients with clinical symptoms.

Among the common co-morbidities of ASD, the prevalence of epilepsy is estimated as 5–38% in children with ASD [130]. Seizures affect 35–60% of patients with mitochondrial diseases [131], which confirms a coexisting pathology. Gastrointestinal dysfunction is another frequent co-morbidity of ASD [132], also seen in mitochondrial disease [100]. Sleep disturbances, especially when associated with the onset and maintenance of sleep, are commonly observed in ASD and can also implicate the behaviour during the daytime, including ASD symptoms [133].

A definitive test for ASD is not given by the DSM-5, the diagnostic reference for ASD. Therefore, a wide range of evaluation tools has been developed for the diagnosis of ASD, e.g., parental questionnaires, parental interviews, clinical judgments, and direct interactions. Of these tests, the Autism Diagnostic Observation Schedule (ADOS) and the Autism Diagnostic Interview-Revised are considered the gold standard for diagnosing ASD [134,135]. The index of daily functionality of an ASD individual should be assessed using adaptative behaviour scales, such as the Vineland Adaptive Behavior Scales (VABS) or Adaptive Behavior Assessment System [136]. However, the nature of the ASD definition, which is primarily based on individual behavioural observation, confounds the diagnosis, especially when an association with an IMD is suspected. Therefore, recent studies have focused on exploring more objective means of understanding the underlying pathogenesis of ASD by investigating specific biomarkers, which would further assist in the diagnosis of any associated IMDs. Thus, a multidisciplinary team of experienced professionals, including clinical psychologists, psychiatrists, paediatricians, and speech and language therapists, is recommended for accurate diagnosis and supportive treatment of ASD [137]. However, access to such services may be limited due to funding shortfalls, especially for adults with undiagnosed ASD, leading to worse functional outcomes. Further, the lack of understanding of the pathophysiology interferes with both diagnostic accuracy and planning for appropriate treatments [8].

Many of the IMDs associated with ASD fall under treatable IMDs, and hence can be managed with targeted (non) pharmacological interventions, reducing morbidity and mortality. For instance, urea cycle disorders can be treated with a protein-defined diet, arginine or citrulline, and nitrogen scavenging agents, whereas cerebral creatine deficiency syndromes can be treated with creatine, glycine, and arginine [138]. A proper clinical assessment is required prior to starting such therapies in order to monitor the response to therapy. The Autism Treatment Evaluation Checklist (ATEC) is a parent-completed assessment tool useful in measuring the effectiveness of various therapeutic interventions [139].

## 5. Conclusions

Metabolic screening is indicated in the diagnostic work-up of children with ASD, especially if they have multiorgan involvement. ASD individuals with IMDs may also exhibit some associated features, such as seizures, gastrointestinal disturbances, and speech problems, which require rapid and specialist intervention. The early recognition and diagnosis of any associated IMDs would increase their likelihood of achieving optimal care and a better quality of life. Furthermore, genetic and epigenetic breakthroughs of underlying pathologies have further paved new avenues to improve the overall health of ASD individuals. The quest for objective biomarkers in ASD and the underlying pathophysiology is still at its preliminary stage, and there is still a vacuum of knowledge yet to be filled with further clinical research to improve the evidence base for various treatment interventions in ASD with associated IMDs. Risk stratification, based on biomarkers with association to IMDs in ASD individuals, would enable improvement of the overall clinical management of ASD.

## Figures and Tables

**Table 1 genes-14-00803-t001:** Inherited metabolic diseases associated with autism spectrum disorder (ASD). All variants were annotated according to the MANE Select transcript. † Disorders with no MIM numbers (AA: amino acid, ADSL: adenylosuccinate lyase, ASLD: argininosuccinate lyase deficiency, CAT-3: cationic amino acid transporter-3, CLTRN: collectrin, CNV: copy number variation, CPS1D: carbamoyl phosphate synthetase 1 deficiency, CTLN1: citrullinemia type I, DHFR: dihydrofolate reductase, GLUT1: glucose transporter 1, MELAS: mitochondrial encephalomyopathy, lactic acidosis and stroke-like episodes, MTHFR: 5,10-methylenetetrahydrofolate reductase, OTCD: ornithine transcarbamylase deficiency, SCADD: short-chain acyl-CoA dehydrogenase deficiency, SSADH: succinic semialdehyde dehydrogenase, SNV: single nucleotide variants, PV: pathogenic or likely pathogenic variant [updated on 11 February 2023 in ClinVar Database or Human Gene Mutation Database], UCDs: urea cycle disorders).

ICIMD Category > ICIMD Subcategory	Inherited Metabolic Disease, Phenotype MIM Number [Ref.]	Gene (MIM Number)/Methods of Genetic Testing
Disorders of AA metabolism > phenylalanine and tyrosine metabolism	Phenylketonuria/phenylalanine hydroxylase deficiency, #261600 [24,33,36,37,38,39,40,41]	*PAH* (*612349); More than 740 PVs have been described. Sanger sequencing analysis is performed first (SNVs = 97–99%), followed by gene-targeted deletion/duplication analysis (1–3%). There are seven common PVs: c.1222C>T (6.7%), c.1066-11G>A (5.3%), c.194T>C (4.1%), c.782G>A (3.6%), c.842C>T (2.9%), c.1315+1G>A (2.8%), c.473G>A (2.7%).
Disorders of AA metabolism > metabolism of sulfur-containing AAs and hydrogen sulfide	Homocystinuria/cystathionine β-synthase deficiency, #236200 [38,41,42]	*CBS* (*613381) Approximately 164 PVs have been described. Sanger sequencing analysis is performed first (SNVs = 95–98%), followed by gene-targeted deletion/duplication analysis (<5%). There are three common PVs: c.833T>C (25% pan-ethnic), c.919G>A (71% in Ireland), c.1006C>T (93% in Qatari).
Disorders of AA metabolism > branched-chain AA metabolism	Branched-chain α-keto acid dehydrogenase kinase deficiency, #614923 [28]	*BCKDK* (*614901) Only eight PVs have been described (SNVs only). Sequence analysis is performed mainly.
	Maple syrup urine disease, #248600 [39]	*DBT* (*248610), *BCKDHB* (*248611), *BCKDHA* (*608348) Approximately 146 in *BCKDHB*, 123 in *BCKDHA,* and 108 in *DBT* of PVs have been described. Sequence analysis is performed first (SNVs 86–93%), followed by gene-targeted deletion/duplication analysis if only one or no pathogenic variant is found (7–14%).
Disorders of AA metabolism > AA transport	Large neutral AA transporter defects ^†^ [43,44]	*SLC3A2* (*158070), *SLC7A5* (*600182), *SLC7A8* (*604235) Some variants are associated with ASD
	Hartnup disease, #234500 [45]	*SLC6A19* (*608893) Only 18 PVs have been described (SNVs only). Sequence analysis is performed mainly.
	CLTRN deficiency ^†^ [46]	*CLTRN* (*300631) A deletion spanning exons 1–3 of *CLTRN* has been reported
	CAT-3 defects ^†^ [47]	*SCL7A3* (*300443) Some variants are associated with ASD
	Lysinuric protein intolerance, #222700 [23]	*SLC7A7* (*603593) Over 90 PVs have been described. Sequence analysis is performed first (SNVs = 92–95%), followed by gene-targeted deletion/duplication analysis (15–20% in the non-Finnish population).
Disorders of AA metabolism > ornithine, proline, and hydroxyproline metabolism	Hyperprolinemia type I, #239500 and type II, #239510 [48]	*PRODH* (*606810) Only 11 PVs have been (only SNVs). Mainly sequence analysis is performed. *ALDH4A1* (*606811) Only seven PVs have been described (SNVs only). Sequence analysis is performed mainly.
Disorders of AA metabolism > glycine and serine metabolism	Serine deficiency disorders [23] Nonketotic hyperglycinemia due to aminomethyltransferase deficiency, #605899 [33]	*AMT* (*238310), *GLDC* (*238300) and *GCSH* (*238330) Approximately 363 in *GLDC*, 88 in *AMT*, and 4 in *GCSH* of PVs have been described. The commonest PV is *GLDC* (80%), *AMT* (20%), and *GCSH* (rare). Sequence analysis is performed first (SNVs = 80% and >99%, respectively), followed by gene-targeted deletion/duplication analysis if only one or no pathogenic variant is found (PV = 20% and unknown, respectively).
Disorders of AA metabolism > UCDs and inherited hyperammonemias	ASLD, #207900 [23,49,50] CTLN1, #215700 [23,41,50] CPS1D, #237300 [24,51] OTCD, #311250 [23,50], unspecified UCDs [51]	*ASL* (*608310) *ASS1* (*603470) *CPS1* (*608307) *OTC* (*300461) Approximately 146 in *ASL,* 141 in *ASS1,* 223 in *CPS1,* and 426 in *OTC* of PVs have been described. SNVs are found in >90% of ASLD cases, 96% of CTLN1 cases, 80–90% of OTCD cases, and >99% of CPS1D cases.
Disorders of AA metabolism > organic acidurias	Propionic acidemia, #606054 [41,52,53]	*PCCA* (*232000), *PCCB* (*232050) Approximately 181 in *PCCA* and 172 in *PCCB* of PVs have been described. Sequence analysis is performed first (SNVs = 78% and >97%, respectively), followed by gene-targeted deletion/duplication analysis if only one or no pathogenic variant is found (CNVs = 3–18%). There are three common PVs in *PCCB*: c.1218_1231del14ins12 (~30% of disease-causing alleles in individuals of northern European origin), c.1304T>C (associated with a milder form of propionic acidemia, accounts for 25% of mutated alleles in Japanese individuals) and c.1606A>G (homozygous in Amish and Mennonite communities, in individuals who can initially present with cardiomyopathy).
	3-methylcrotonyl-coA carboxylase deficiency, #210200 [41]	*MCCC1* (*609010) Approximately 119 PVs have been described (SNVs > 99%). Sequence analysis is performed mainly.
	Isovaleryl-coA dehydrogenase deficiency, #243500 [41,54]	*IVD* (*607036) Approximately 119 PVs have been described (SNVs = 98%). Sequence analysis is performed mainly.
	2-methylbutyrylglycinuria, #610006 [49]	*ACADSB* (*600301) Only 16 PVs have been described (SNVs = 94%). Sequence analysis is performed mainly.
	Glutaric aciduria type 1, #231670 [40]	*GCDH* (*608801) Approximately 223 PVs have been described. Sequence analysis is performed mainly (SNVs = 99%).
Disorders of AA metabolism > other disorders of AA metabolism	Aminoacylase 1 deficiency, #609924 [55]	*ACY1* (*104620) Only 11 PVs have been described (SNV only). Sequence analysis is performed mainly.
Disorders of metabolite repair/proofreading > mitochondrial metabolite repair	L-2-hydroxyglutaric aciduria, #236792 [56]	*L2HGDH* (*609584) Only 30 PVs have been described (SNVs > 99%). Sequence analysis is performed mainly.
	Combined malonic and methylmalonic aciduria, #614265 [49]	*ACSF3* (*614245) More than 100 PVs have been described (SNVs only). Sequence analysis is performed mainly.
Disorders of fatty acid and ketone body metabolism > mitochondrial fatty acid oxidation	SCADD, #201470 [42,49]	*ACADS* (*606885) Approximately 70 PVs have been described (SNVs only). Sanger sequencing analysis is performed first. There are two common PVs: c.511C>Tandc.625G>A, which result in the SCADD biochemical abnormality when in trans with an apathogenic variant. Newborns homozygous for the c.625G>A variant have laboratory test values that overlap with those of affected newborns.
Disorders of fatty acid and ketone body metabolism > carnitine metabolism	Primary carnitine deficiency, #212140 [41]	*SLC22A5* (*603377) More than 189 PVs have been described (SNVs > 99). Sequence analysis is performed mainly.
	Epsilon-n-trimethyllysine hydroxylase deficiency, #300872 [27]	*TMLHE* (*300777) Only three PVs have been described (SNVs only). Sequence analysis is performed mainly.
Disorders of fatty acid and ketone bodies > ketone body metabolism	Mitochondrial acetoacetyl-CoA thiolase deficiency, #203750 [54]	*ACAT1* (*607809) More than 150 PVs have been described (SNVs = 96%). Sequence analysis is performed mainly.
Disorders of vitamin and cofactor metabolism > biotin metabolism	Biotinidase deficiency, #253260 [23] Partial biotinidase deficiency [39,42]	*BTD* (*609019) Approximately 145 PVs have been described (SNVs only). Sequence analysis is performed mainly.
Disorders of vitamin and cofactor metabolism > folate metabolism	Low-activity variants of MTHFR [57,58]	*MTHFR* (*607093) Approximately 117 PVs have been described (SNVs only). Sequence analysis is performed mainly. Thermolabile low-activity c.677C>T variant is associated with ASD
	Folate receptor α deficiency, #613068 [23]	*FOLR1* (*136430) Only 21 PVs have been described (SNVs only). Sequence analysis is performed mainly.
	Low-activity variants of DHFR [58]	*DHFR* (*126060) Only two PVs have been described (SNVs only). Sequence analysis is performed mainly
Disorders of vitamin and cofactor metabolism > molybdenum cofactor metabolism	Molybdenum cofactor deficiency, #252150 [59,60]	*MOCS1* (*603707) Only 32 PVs have been described (SNVs > 99%). Sequence analysis is performed mainly
Disorders of vitamin and cofactor metabolism > cobalamin metabolism	Transcobalamin II deficiency, #275350 [61]	*TCN2* (*613441) Only 29 PVs have been described (SNVs > 93%). Sequence analysis is performed mainly.
Disorders of lipid metabolism > sterol biosynthesis	Smith–Lemli–Opitz syndrome, #270400 [62,63]	*DHCR7* (*602858) More than 180 PVs have been described (SNVs = 96%). Sanger sequencing analysis is performed first. There are three common variants: c.964-1G>C (~28% of disease alleles; associated with a severe phenotype), c.452G>A (the most common in Central Europe; associated with a severe phenotype), c.278C>T (common in individuals of Mediterranean or Cuban ancestry; associated with a milder phenotype).
Disorders of lipid metabolism > bile acid metabolism	Cerebrotendinous xanthomatosis, #213700 [64]	*CYP27A1* (*606530) More than 140 PVs have been described (SNVs > 99%). Sanger sequencing analysis is performed first. There are three common variants: c.355delC (founder variant in Israeli Druze) and c.1183C>T.
Disorders of complex molecule > glycosaminoglycan degradation	Sanfilippo syndrome, #252920 [23,38,39,42,65]	*NAGLU* (*609701) More than 200 PVs have been described (SNVs >90%). Sequence analysis is performed mainly.
	Morquio syndrome A, #253000 [39]	*GALNS* (*612222) Approximately 209 PVs have been described (SNVs > 94%). Sequence analysis is performed mainly. One of the most common PVs is c.1156C>T, accounting for 8.9%.
Disorders of carbohydrate metabolism > carbohydrate transmembrane transport and absorption	GLUT1 deficiency, #612126 [23,66]	*SLC2A1* (*138140) More than 250 PVs have been described (SNVs = 84%). Sequence analysis is performed mainly. Several pathogenic hot spots have been detected: c.376C>T, c.377G>A, c.377G>T, c.766_767delAAinsGT, c.997C>T, c.884C>T, c.1402C>T.
Disorders of carbohydrate metabolism > glycogen metabolism	Glycogen storage disease type IXa1, #306000 [67]	*PHKA2* (*300798) More than 250 PVs have been described (SNVs = 94%). Sequence analysis is performed mainly.
mtDNA-related disorders > disorders of mtDNA-encoded tRNA and rRNA > disorders of mtDNA-encoded oxidative phosphorylation proteins	MELAS syndrome, #540000 [68,69]	Several genes: *MTTL1* (*590050), *MTTQ* (*590030), *MTTH* (*590040), *MTTK* (*590060), *MTTC* (*590020), *MTTS1* (*590080), *MTND1* (*516000), *MTND5* (*516005), *MTND6* (*516006), and *MTTS2* (*590085) More than 90 PVs have been described (SNVs only). Proportion PVs are *MTTL1* (>80%), *MTND5* (<10%), and other genes (rare). Entire mitochondrial genome sequencing that includes *MTTL1*, *MTND5*, and other mtDNA genes is most effective in identifying the genetic cause of the condition.
Neurotransmitter disorders > γ-aminobutyric acid neurotransmitter disorders	SSADH deficiency, #271980 [23,24,67,70]	*ALDH5A1* (*610045) Approximately 108 PVs have been described (SNVs = 97%). Sequence analysis is performed mainly.
Disorders of nucleobase, nucleotide, and nucleic acid metabolism > purine metabolism	ADSL deficiency, #103050 [23,71] Lesch–Nyhan syndrome, #300322 [23,24]	*ADSL* (*608222) Approximately 58 PVs have been described (SNVs only). Sequence analysis is performed mainly. *HPRT1* (*308000) Approximately 87 PVs have been described. Sequence analysis is performed first (SNVs = 80%), followed by gene-targeted deletion/duplication analysis (CNVs = 20%).
Disorders of nucleobase, nucleotide, and nucleic acid metabolism > pyrimidine metabolism	Dihydropyrimidine dehydrogenase deficiency, #274270 [23,67]	*DPYD* (*612779) Approximately 50 PVs have been described (SNVs only). Sequence analysis is performed mainly.
Disorders of energy substrate metabolism > creatine metabolism	Cerebral creatine deficiency syndromes, #300352 [23,39,49,67,72,73,74]	*SLC6A8* (*300036), *GAMT* (*601240), *GATM* (*602360) Approximately 100 in *SLC6A8*, 73 in *GAMT* and 17 in *GATM* of PVs have been described (SNVs 95–100%). The proportion of PVs is *SLC6A8* (64–72%), *GAMT* (20–33%), and *GATM* (3–8%). Sequence analysis is performed mainly.

**Table 2 genes-14-00803-t002:** Basic screening [100,101,102,103,104,105,106,107,108,109,110,111,112]. (3-MCCD: 3-methylcrotonyl-CoA carboxylase deficiency, CTX: cerebrotendinous xanthomatosis, GSD IX: glycogen storage disorder IX, MTHFRD: methylenetetrahydrofolate reductase deficiency, MSUD: maple syrup urine disease, NKH: nonketotic hyperglycinemia, PA: propionic acidemia, PKU: phenylketonuria, SCADD: short-chain acyl-CoA dehydrogenase deficiency).

Test Category	Investigation	Findings in Different IEM
Biochemical	Plasma glucose ^a^	Hypoglycemia in 3-MCCD, GSD IX, PA, primary carnitine deficiency, SCADD
	Serum liver transaminases	Increased in GSD IX, lysinuric protein intolerance, primary carnitine deficiency, SCADD, urea cycle disorders, Wilson’s disease
	Serum lactate dehydrogenase	Increased in lysinuric protein intolerance
	Serum uric acid	Increased in Lesch–Nyhan syndrome
		Decreased in molybdenum cofactor deficiency, Wilson’s disease
	Serum urea	Decreased in urea cycle disorders, lysinuric protein intolerance
	Serum cholesterol	Decreased in Smith–Lemli–Opitz syndrome, CTX
		Increased ^b^ in GSD IX, lysinuric protein intolerance
	Serum triglyceride	Increased in GSD IX, lysinuric protein intolerance
	Acid-base status	Metabolic acidosis with high anion gap in biotinidase deficiency, MSUD, organic acidurias, primary carnitine deficiency, SCADD
		Respiratory alkalosis in urea cycle disorders
	Serum/urine ketones	Increased in 3-MCCD, biotinidase deficiency, GSD IX, isovaleryl-CoA dehydrogenase deficiency, MSUD, PA
Haematological	WBC	Neutropenia in isovaleryl-CoA dehydrogenase deficiency, lysinuric protein intolerance, PA, Wilson’s disease
		Vacuolated lymphocytes on peripheral blood smear in Morquio syndrome A, Sanfilippo syndrome
	Platelets	Thrombocytopenia in 3-MCCD, isovaleryl-CoA dehydrogenase deficiency, lysinuric protein intolerance, PA
	RBC	Anemia in isovaleryl-CoA dehydrogenase deficiency, lysinuric protein intolerance, PA, Wilson’s disease
Preliminary urine screening tests	Ferric chloride test ^c^	Produce an intense blue-green in the presence of phenylpyruvate in PKU, MSUD
	Cyanide-nitroprusside test with silver nitrate ^c^	Produce a magenta colour in the presence of homocysteine in homocystinuria
	2,4-dinitrophenylhydrazine test ^d^	Orange precipitate in MSUD
	Sulfite test ^e^	Positive in molybdenum cofactor deficiency
	Glycosaminoglycan/creatinine ratio (DMMB assay)	Increased in mucopolysaccharidoses (e.g., Sanfilippo syndrome)
Dried blood spot (DBS) screening	Phenylalanine ^f^	Increased in PKU
	Hypermethioninemia ^g^	Increased in homocystinuria
	Leucine + isoleucine	Increased (leucine + isoleucine) to alanine and phenylalanine in MSUD
Radiological studies	X-ray	Osteoporosis in PA
		Dysostosis multiplex in Morquio syndrome A
	MRI	White matter involvement with atrophic cortical changes in PKU, 3-MCCD, CTX, ADSL deficiency
EEG		Abnormalities in 3-MCCD, biotinidase deficiency, CTX, Folate receptor α deficiency, NKH, PKU

Special notes [105,109,110,112]. ^a^ Hypoglycaemia may not be present if the patient is not in metabolic decompensation. ^b^ Cholesterol can be normal. ^c^ False-negative results may be seen in dilute urine specimens. ^d^ Test detects α-ketoacids formed from the isoleucine, leucine, and valine. ^e^ Due to the poor stability of urinary sulfite, drugs, and bacterial degradation, it is prone to false-negative results. ^f^ Assayed by TD-MS/MS. Diurnal variability in phenylalanine concentrations may necessitate repeat screening for PKU. ^g^ Hypermethioninemia may not always be present in the neonatal period.

**Table 3 genes-14-00803-t003:** Metabolic work-up [100,101,102,103,104,105,106,107,108]. (3-MCCD: 3-methylcrotonyl-CoA carboxylase deficiency, AA: amino acid, AIP: acute intermittent porphyria, CSF: cerebrospinal fluid, CTX: cerebrotendinous xanthomatosis, GSD IX: glycogen storage disorder IX, MTHFRD: methylenetetrahydrofolate reductase deficiency, MSUD: maple syrup urine disease, NKH: nonketotic hyperglycinemia, OCT deficiency: ornithine transcarbamylase deficiency, PA: propionic acidemia, PKU: phenylketonuria, SCADD: short-chain acyl-CoA dehydrogenase deficiency, SSADH: succinic semialdehyde dehydrogenase).

Test Category	Investigation	Finding	IEM
Blood metabolites	Plasma lactate ^a^	Increased	Biotinidase deficiency, GSD IX, PA
	Plasma ammonia ^b^	Increased	3-MCCD, biotinidase deficiency, PA, urea cycle disorders
	Serum uric acid	Decreased	Molybdenum cofactor deficiency
	Plasma free fatty acid ^c^	Increased	SCADD
	Serum sterol analysis (7-dehydrocholesterol, 8-dehydrocholesterol)	Increased	Smith–Lemli–Opitz syndrome
	Serum copper (Cu)	Increased	Wilson’s disease
	Serum ceruloplasmin ^d^	Decreased	Wilson’s disease
	Serum transcobalamin	Decreased	Transcobalamin II deficiency
	Plasma succinylaminoimidazole carboxamide riboside, succinyladenosine	Increased	Adenylosuccinate lyase deficiency
Blood vitamin levels	Serum folate	Normal	Folate receptor α deficiency
	Serum cobalamin (Vit B12)	Normal	Transcobalamin II deficiency
Plasma AA ^e^	Phenylalanine	Increased	PKU
	Methionine	Increased	Homocystinuria
	Homocysteine	Increased	Homocystinuria, MTHFRD, transcobalamin II deficiency
	Cysteine	Increased	Homocystinuria
		Decreased	Molybdenum cofactor deficiency
	S-sulfocysteine	Increased	Molybdenum cofactor deficiency
	Taurine	Increased	Molybdenum cofactor deficiency
	Leucine, isoleucine, alloisoleucine	Increased	MSUD
	Neutral AA (Alanine, serine, threonine, asparagine, glutamine, valine, leucine, isoleucine, phenylalanine, tyrosine, tryptophan, histidine, citrulline)	Decreased	Hartnup disorder, collectrin deficiency
	Proline	Increased	Hyperprolinemia
	Glycine	Increased	NKH-AD, SSADH
	Arginine	Decreased	Argininosuccinate lyase deficiency, citrullinemia, OTC deficiency
	Citrulline	Increased	Citrullinemia, argininosuccinate lyase deficiency
		Decreased	OCT deficiency
	Proline	Increased	Citrullinemia, OCT deficiency
	Isovaleric acid	Increased	Isovaleryl-CoA dehydrogenase deficiency
CSF AA ^f^	Phenylalanine	Increased	PKU
	Glycine	Increased	NKH-AD, SSADH
	Homocysteine	Increased	Folate receptor α deficiency MTFRD
	Methionine	Decreased	MTFRD
CSF analytes	Glucose, lactate	Decreased	GLUT1 deficiency
	Folate	Decreased	Folate receptor α deficiency
	5-methyl THF	Decreased	MTHFRD
	Succinylaminoimidazole carboxamide riboside, succinyladenosine	Increased	ADSL deficiency
Plasma carnitine profile ^g^	Propionyl carnitine	Increased	Biotinidase deficiency, PA
	3-hydroxyisovaleryl (CS-OH)	Increased	3-MCCD, Biotinidase deficiency
	Carnitine; esterified	Increased	3-MCCD
	Carnitine; total and free	Decreased	3-MCCD
	Butyrylcarnitine	Increased	SCADD
	Free carnitine	Decreased	SCADD
Plasma porphyrins	ALA, PBG, UPIII	Increased	AIP
Urinary metabolites	ALA, PBG	Increased	AIP
	Bile acids	Increased	CTX
	Pterins	Increased	PKU
	Xanthine and hypoxanthine, thiosulfate	Increased	Molybdenum cofactor deficiency
	Sulfate	Decreased	Molybdenum cofactor deficiency
Urine organic acid profile ^h^	Phenylpyruvate	Increased	PKU
	Branched-chain α-ketoacids	Increased	MSUD
	Argininosuccinate	Increased	Argininosuccinate lyase deficiency
	Orotic acid	Increased	Argininosuccinate lyase deficiency, citrullinemia, Lysinuric protein intolerance
	3-hydroxypropionate, methyicitrate, propionylglycine	Increased	Propionic acidemia
	3-hydroxyisovaleric acid, 3-methylcrotonylglycine	Increased	3-MCCD
	Iso-valerylglycine, 3-OH-isovaleric acid	Increased	Isovaleryl-CoA dehydrogenase deficiency
	Glutaric acid	Increased	Glutaric aciduria
	Ethylmalonate, methylsuccinate, butyrylglycine	Increased	SCADD
	γ-Hydroxybutyric acid	Increased	SSADH
	Cystathionine	Increased	MTHFRD
	Methyimalonic acid	Increased	Transcobalamin II deficiency
Urine AA	Phenylalanine	Increased	PKU
	Homocysteine	Increased	Homocystinuria Transcobalamin II deficiency
	S-sulfocysteine	Increased	Molybdenum cofactor deficiency
	Taurine	Increased	Molybdenum cofactor deficiency
	Leucine + isoleucine	Increased	MSUD
	Neutral AA (Alanine, serine, threonine, asparagine, glutamine, valine, leucine, isoleucine, phenylalanine, tyrosine, tryptophan, histidine, and citrulline)	Increased	Hartnup disorder
	Hydroxyproline, glycine	Increased	Hyperprolinemia
	Citrulline	Increased	Argininosuccinate lyase deficiency
	Homocitrulline	Increased	Citrullinemia OTC deficiency
	Glycine	Increased	Propionic acidemia
Urine mucopolysaccharide excretion profile ^i^	Heparan sulfate	Increased	Sanfilippo syndrome
	Keratan sulfate	Increased	Morquio syndrome A
Stool amino acid	Neutral AA (Alanine, serine, threonine, asparagine, glutamine, valine, leucine, isoleucine, phenylalanine, tyrosine, tryptophan, histidine, and citrulline)	Increased	Hartnup disorder Collectrin deficiency
Enzyme activity	Propionyl-CoA carboxylase activity (WBC, fibroblasts)	Decreased	Propionic acidemia
	Carboxylase activity (WBC) Biotinidase activity	Decreased	Biotinidase deficiency
	27-hydroxylase (Fibroblasts)	Decreased	CTX
	Glycogen phosphorylase kinase (Lymphocytes, RBC)	Decreased	GSD IX
	N-acetylglucosamine 6-sulfatase (Fibroblasts)	Decreased	Sanfilippo syndrome
	N-acetylgalactosamine 6-sulfatase	Decreased	Morquio syndrome A
	Sphingomyelinase (WBC, fibroblasts)	Decreased	Niemann–Pick disease

Special notes [105,112,113]. ^a^ Specimen: NaF/oxalate tube, immediately transported to lab on ice. ^b^ Specimen: heparin tube, immediately transported to the lab on ice and refrigerated centrifugation to separate plasma. Hyperammonemia may not be present if the patient is not in metabolic decompensation. ^c^ Specimen: NaF/oxalate tube, store separated plasma frozen. ^d^ Being a positive acute-phase reactant, false elevation can be seen in infection/acute hepatitis. Inadequate dietary copper intake and malabsorption can lead to low levels. ^e^ Assayed by HPLC. Best done during an acute episode. Patient preparation for non-acute samples: ideally collected after at least a 3-h fast or immediately before the next feed. Specimen: heparin tube on ice, plasma separated within 1 h of collection, and stored frozen. Interferences: post-prandial samples may show spuriously increased amino acid levels. ^f^ Assayed by HPLC. No patient preparation is indicated. Specimen: clear CSF, snap frozen on dry ice, and stored frozen. Interferences: results are uninterpretable from blood-stained samples. Blood-stained specimens can be centrifuged to obtain clear supernatant. ^g^ Specimen: heparin tube, stored separated plasma frozen. ^h^ Assayed by GC-MS. No patient preparation is indicated. Specimen: mid-stream urine, stored frozen. Urinary excretion of BCKD was noted 48 h after birth in infants on an unrestricted diet. ^i^ Assayed by LC-MS/MS. Specimen: early morning mid-stream urine. Dilute urines may yield uninterpretable profiles.

## Data Availability

No raw data is available for this review.

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
