# Peer review of "Biochemical, Genetic and Clinical Diagnostic Approaches to Autism-Associated Inherited Metabolic Disorders"

_genes, 2023, doi:10.3390/genes14040803_

Round 1

Reviewer 1 Report

This is a complete review of inherited metabolic conditions presenting with austism spectrum disorder features. The manuscript suggests to consider these subgroup of ASD conditions when facing a possible diagnosis and to use accurately information emerging from extended neonatal screening.

The text is well presented and complete. Minor changes should indicate

- the possibility to have confirmatory biochemical evidences of gene variants

- the potential treatments and how they can impact on ASD features

Author Response

Reviewer 1 Comments

  1. the possibility of having confirmatory biochemical evidence of gene variants

AUTHOR RESPONSE: We thank you for this valuable comment. This was included in the text.

  1. the potential treatments and how they can impact ASD features

RESPONSE: many thanks for the suggestion, even though the potential of the therapies is not an objective we have outlined the significance of diagnosis and mode of monitoring symptoms using ATEC, under “Clinical and neuropsychology assessments”

Reviewer 2 Report

The review comprehensively describes the possible implications of inherited metabolic disorders in autism. Manuscript develops the correlation by summarizing the information in a comprehensive table in which they identify the disorder, phenotype, and possible candidate genes highlighting biochemical features.

Additional diagnostic biomarkers, such as cytokines, that play a role in modulating the gut-brain axis, should be commented. Indeed, it is reported in the literature that microbial communities play a central role in the maturation and development of the immune system, central nervous system, and gastrointestinal system and are also responsible for essential metabolic pathways. An imbalance in the gut microflora is associated with the onset and progression of neurological disorders, such as ASD.

In this review, the authors mention the correlation between the folate cycle and autism, so in addition to the CBS gene, which they also report in table 1, they could report other important folate genes, such as MTHFR, DHFR.

In the section on genetic investigation, the authors highlight the importance of detecting alterations in non-coding sequences of the genome. In this context, it is crucial to describe the relationship between epigenetics and autism. Several scientific studies report that epigenetic modifications such as methylation represent risk factors for ASD.

Finally, sex and gender analyses are mandatory in ASD

Minor

1.      Table 1, page 2:  references for the CBS gene. References n° 39 and 42 are not perfectly appropriate

2.  Paragraph “Biochemical investigations, line 3: the reference partially describes what is stated

3.      Page 13, paragraph “Genetic investigations, lines 1- 7: please add references

4.      Page 14, line 10: Please rephrase “some centres also detect alterations in the remainder of gene encoding..”

Suggested recent references to be included:

- Jensen AR, et al. Modern Biomarkers for Autism Spectrum Disorder: Future Directions.Mol Diagn Ther. 2022 Sep;26(5):483-495. doi: 10.1007/s40291-022-00600-7.

- Sorboni SG, et al. Comprehensive Review on the Role of the Gut Microbiome in Human Neurological Disorders. Clin Microbiol Rev. 2022 Jan 19;35(1):e0033820. doi: 10.1128/CMR.00338-20.

- Tisato V, et al. Genetics and Epigenetics of One-Carbon Metabolism Pathway in Autism Spectrum Disorder: A Sex-Specific Brain Epigenome? Genes. 2021 May 20;12(5):782. doi: 10.3390/genes12050782.

Author Response

Reviewer 2 Comments

  1. Additional diagnostic biomarkers, such as cytokines, that play a role in modulating the gut-brain axis, should be commented.

RESPONSE: We thank you for this valuable comment. We discussed its significance briefly in the introduction. However, we did not discuss this in-depth as not relevant to inborn errors associated with ASD.

  1. Additional diagnostic biomarkers, such as cytokines, that play a role in modulating the gut-brain axis, should be commented. Indeed, it is reported in the literature that microbial communities play a central role in the maturation and development of the immune system, central nervous system, and gastrointestinal system and are also responsible for essential metabolic pathways. An imbalance in the gut microflora is associated with the onset and progression of neurological disorders, such as ASD.

RESPONSE: We thank you for this valuable comment. We discussed this fact briefly in the introduction. However, we did not discuss this in-depth as not relevant to inborn errors associated with ASD.

  1. In this review, the authors mention the correlation between the folate cycle and autism, so in addition to the CBS gene, which they also report in table 1, they could report other important folate genes, such as MTHFR, and

RESPONSE: We thank you for this valuable comment. We have included MTHFR and DHFR variants to the table.

  1. In the section on genetic investigation, the authors highlight the importance of detecting alterations in non-coding sequences of the genome. In this context, it is crucial to describe the relationship between epigenetics and autism. Several scientific studies report that epigenetic modifications such as methylation represent risk factors for ASD.

RESPONSE: We thank you for this valuable comment. This is now addressed in the manuscript.

Recent studies have also pointed toward the role of epigenetic dysregulation, mechanisms with no impact on the DNA sequence, in the ASD pathogenesis (121). Epigenetics can be defined as the regulation of gene expression by modulating chromatin formation (122), and reported instances of such dysregulations include, impaired DNA methylation in ASD individuals (123, 124), mutation in the HIST1H1E gene encoding H1 histone linker protein (125), activating immune responses during pregnancy increasing susceptibility to ASD (126), alteration of the micro RNA expression in ASD patients (127).

  1. Finally, sex and gender analyses are mandatory in ASD

RESPONSE: We thank you for this valuable comment. This is now addressed in the manuscript.

A gender bias has been observed historically in the incidence of neurodevelopmental disorders, with a 4-5-fold higher incidence in males, which is more prominent in individuals with a milder degree of cognitive impairment (4, 5). Although this phenome-non would point toward the possibility of an association with X-linked genes, only 10% of the reported male excess can be attributed to X-linked genes. Further, ASD recurrent rates among siblings of a proband are estimated to be 4% and 7%, for males and females, respectively, with the less frequent gender exhibiting a higher recurrence rate (6, 7).

  1. Table 1, page 2:  references for the CBS gene. References n° 39 and 42 are not perfectly appropriate

      RESPONSE: We thank you for this valuable comment. Those references were removed.

  1. Paragraph “Biochemical investigations, line 3: the reference partially describes what is stated

RESPONSE: We thank you for this valuable comment. The reference was edited as suggested.

  1. Page 13, paragraph “Genetic investigations, lines 1- 7: please add references

RESPONSE: We thank you for this suggestion. The references were added and updated.

  1. Page 14, line 10: Please rephrase “some centres also detect alterations in the remainder of gene encoding..”

RESPONSE: We thank you for this valuable comment. This was included in the text.

  1. Suggested recent references to be included:

- Jensen AR, et al. Modern Biomarkers for Autism Spectrum Disorder: Future Directions.Mol Diagn Ther. 2022 Sep;26(5):483-495. doi: 10.1007/s40291-022-00600-7.

RESPONSE: We thank you for this valuable comment. This reference was included in the introduction with some additions to the text.

- Sorboni SG, et al. Comprehensive Review on the Role of the Gut Microbiome in Human Neurological Disorders. Clin Microbiol Rev. 2022 Jan 19;35(1):e0033820. doi: 10.1128/CMR.00338-20.

RESPONSE: We thank you for this valuable comment. This reference was included in the introduction with some additions to the text.

- Tisato V, et al. Genetics and Epigenetics of One-Carbon Metabolism Pathway in Autism Spectrum Disorder: A Sex-Specific Brain Epigenome? Genes. 2021 May 20;12(5):782. doi: 10.3390/genes12050782.

RESPONSE: We thank you for this valuable comment. This reference was included in the table and “biochemical investigations” section.

Reviewer 3 Report

Dear authors,

I have only a few suggestions:

- the tables are large and difficult to follow, a header should be inserted for each page to make the table easier to read;

- tables 2 and 3 are missing references.

Author Response

Reviewer 3 Comments

  1. the tables are large and difficult to follow, a header should be inserted for each page to make the table easier to read;

RESPONSE: We thank you for this valuable comment. Information on less frequently reported disorders was removed from the tables. Historical tests such as the Berry spot test are mundane and associated with high false positive rates, hence removed. When appropriate, the disease group was used in the table so that all the disorders falling under such group need not be mentioned. Also, a header was inserted for each page when the table continued to the next page for easy reference.

  1. tables 2 and 3 are missing references.

RESPONSE: We thank you for this valuable comment. Referenced were included in the table legend.

Round 2

Reviewer 1 Report

none